# Using unofficial media, less trusting of Chinese polity?—An analysis based on the moderated mediation effect

**Qian Hu** *, **Yanping Pu**

School of Public Policy and Administration, Chongqing University, Chongqing, China

* huqian@cqu.edu.cn

## Abstract

Enhancing political trust is an important manifestation of China's ability to modernisation national governance in the media age. In the context where unofficial media has a crowding-out effect on official media, building political trust effectively becomes an important foundation for promoting the construction of a national governance system. This study employs the 2015 survey data on social consciousness of netizens and constructs a moderated mediation model using the bootstrap method, with subjective well-being as the intermediary variable and official media use as the moderating variable, to empirically explore the influence of unofficial media use on political trust and its underlying mechanism. The results reveal that unofficial media use significantly and steadily deconstructing political trust. In terms of the mechanism of transmission, subjective well-being is an important channel used by unofficial media use to deconstruct political trust, official media has a positive moderating role in the impact pathway of subjective well-being on political trust. Further research finds that unofficial media use has a stronger impact on trust in the central government, court, and police, compared to trust in township governments. Weibo or online communities and overseas media can deconstruct political trust, however, gossip or chatting with friends can construct political trust. In general, this study provides theoretical basis and empirical experience for how to enhance government trust and ultimately promote the construction of national governance system, given the increasing influence of unofficial media. Meanwhile, the research results also provide some reference value for countries with similar backgrounds to China.

## Introduction

Many studies have shown that the regime may be overthrown due to a lack of trust [1, 2], political trust is closely related to the legitimacy of the regime and political stability, and is regarded as an important feature of democracy [3–6], therefore, maintaining political trust has always been one of the primary tasks of the government [7]. Political trust is divided into narrow sense and broad sense. Political trust in the narrow sense is equivalent to government trust [8], that is, the degree of public trust in the administrative system; Political trust in a broad sense refers to the degree of public trust in polity, political organizations, state institutions and

**Data Availability Statement:** All relevant data are within the paper and its Supporting information files.

**Funding:** We are grateful for the financial support from 1.National Social Science Fund of China (Project No.20XJY004), the grant was awarded to

Yanping Pu, who was mainly involved in funding acquisition, supervision, and writing-review & editing. 2.Central University Basic Research Fund of China (Project No.2020CDJSK01TD02), the grant was awarded to Yanping Pu, who was mainly involved in funding acquisition, supervision, and writing-review & editing. 3.Scientific Research Plan Projects of Shaanxi Education Department (Project No.21JK0148), the grant was awarded to Ran Gu, who had no role in study design, data collection and analysis, decision to publish, or preparation of the manuscript.

**Competing interests:** The authors have declared that no competing interests exist.

government institutions [9]. The formation of political trust is a relatively complicated process. Institutional perspective and cultural perspective are often used to explain the generation mechanism of political trust [10]. Institutional perspective emphasizes that political trust is endogenous [11], government performance [12], social policy, political efficacy [13] and other factors will affect political trust. Cultural perspective emphasizes that political trust is exogenous [11], and political trust is affected by traditional values [14, 15] and social capital. Although institutional perspective and cultural perspective have their own focus, they are not mutually exclusive, but complementary [8]. Institutional perspective emphasizes whether the performance of the object of political trust is trustworthy, and discusses the problem of 'asking me to believe'. Cultural perspective, highlights the main feature of political trust, that is, whether the public has the tendency to give trust, and studies the problem of 'I want to believe'. No matter exploring the question of 'asking me to believe' or 'I want to believe', we should not underestimate the crucial role of media as a bridge between the subject and object of political trust. Especially in recent years, the rapid development of network media technology has provided new avenues for explaining political trust. The advantage of media in explaining political trust lies in its emphasis on information dissemination as an important source of public performance perception, compared to institutionalism; and in highlighting the shaping effect of media on public values, compared to culturalism.

The media can have an impact on political trust, mainly due to the functions of media dissemination and cultivation, as well as the existence of ideology. In this era of rapidly evolving information, the public is constantly surrounded by various types of media, and their evaluation of political trust is not solely based on direct contact and experience, but often comes from media dissemination. Due to the different political culture and media context, media is regarded as the fourth political power in western countries [16], while in China, it is considered as a tool for political propaganda [17], plays an important role in maintaining the stability of the regime and strengthening social governance [18]. In Western democracies, the media has a significant impact on the rise and fall of political trust that have been confirmed [19, 20]. However, is this true for China as well? The answer is inconclusive [21–25]. The main reason is that scholars focus only on a specific type of media or a specific context [26]. Previous researches have often explored the impact of traditional media use and new media use on political trust based on the media's communication technology perspective. Studies have generally found that traditional media use significantly constructs political trust [27, 28], new media use significantly deconstructs political trust [29], but research lacks a media organization perspective to examine the impact of official media use and unofficial media use on political trust.

According to media's political affiliation, media can be classified as official and unofficial [30]. While the development of internet technology has provided a strong support for the growth of media, it has also brought about the emergence of unofficial media, which undermines the government's monopoly on discourse. This is mainly manifested in the 'crowding-out effect' of unofficial media on official media, which weakens the public's reliance on official media and correspondingly increases their exposure to unofficial media and the possibility of receiving negative political news. Thus, in the context of the crowding effect of unofficial media on official media, it is of great practical significance to examine the impact of attention to unofficial media on government trust and its mechanisms. At the same time, this can provide empirical evidence for advancing the national governance system and provide some reference value for countries with similar backgrounds to China.

The existing research literature on the relationship between unofficial media use and political trust has the following limitations: firstly, previous studies on the impact of political trust have mainly explained it from an institutionalist and culturalist perspective, neglecting to

explain political trust from a media perspective. Secondly, the literature mainly investigates the direct effect of unofficial media use on political trust, neglecting the complexity of the mechanisms underlying the formation of political trust. In fact, unofficial media use can also indirectly affect political trust through other mediating variables, or be subject to the moderating effects of other variables that lead to heterogeneity in the impact on political trust. Lastly, the existing literature mainly investigates the impact of unofficial media use on political trust from a holistic perspective, lacking a nuanced analysis of the horizontal dimensions that refine the content of the relationship between the use of unofficial media and political trust. Moreover, very little attention has been paid to the endogeneity issues caused by selection bias.

In view of this, this study systematically investigates the influence mechanism of unofficial media use on political trust through the data of 2015 netizens' social awareness survey. The main contributions of this paper to existing research are as follows. First, from a media perspective, this study establishes and validates a mechanism for how unofficial media use impacts political trust through deductive reasoning. It also enhances and broadens the existing research findings on the determinants of political trust. Second, the present study aims to examine the role of subjective well-being as an intermediary variable, independent of institutional and cultural factors. In contrast to prior research, this paper does not 'isolation' or 'merge' unofficial media use with official media use, but instead employs a moderating effect approach to investigate the influence of official media use on the mediating relationship between subjective well-being and political trust. This approach provides deeper insight into the underlying reasons for the impact of unofficial media use on political trust. Third, building on the broader investigation into the mechanisms underlying the impact of unofficial media use on political trust, this study delves deeper into the horizontal dimension by examining the specific effects of different types of unofficial media on trust in distinct political institutions. This approach refines and expands the scope of research on political trust, providing empirical evidence to better understand the influence of unofficial media use on political trust.

## Literature review and theoretical hypothesis

### Unofficial media use and political trust

The theory of socialization places great emphasis on the role of media in shaping the political attitudes of netizens. Through the function of agenda-setting, selective reporting, language choice, and moral evaluation, media can influence netizens' attitudes and behaviors on a specific issue, and even modify their views to align with the dominant perspective presented by the media. Thus, media use has a significant impact on netizens' level of political trust and can influence their real-time political information processing and decision-making. Within the field of political communication, there are currently two prevailing theories that examine the connection between media and political trust. The theory of media depression emphasizes that the public's media use content often results in adverse political attitudes, including political apathy, dissatisfaction, and distrust [31]. This phenomenon occurs because media outlets tend to prioritize sensational and negative news stories, such as corruption, environmental degradation, and political scandals, which can evoke feelings of anger and discontent among the public and erode their political trust [32, 33]. In other words, the constant exposure to negative media content can contribute to a pessimistic outlook on politics. The benign-cycle theory suggests that the public's media use can have positive effects on political attitudes, such as increasing political interest, knowledge, and participation, as well as fostering political trust [34–37]. However, research indicates that the relationship between media use and political trust is complex and dependent on various factors, including the type and

content of media consumed, the level of media use and trust, and the underlying motivation for media consumption [24, 38, 39]. Additionally, the impact of media use on political trust is subject to change depending on the broader media landscape and context [40, 41]. Given this, it is worth examining how unofficial media use affects political trust.

Scholars have observed that unofficial media use can undermine political trust [35, 42, 43]. Public's demand for diversified information acquisition and official media's silence on political scandals, environmental pollution, social injustice and other issues prompt the public to seek unofficial media as information sources [7]. Unofficial media outlets are often marked by their lack of structure, openness, and a focus on market-oriented benefits rather than government control. This environment can encourage the reporting of dramatic and sensational information, particularly in the realm of negative political news, meanwhile, public also prefers to focus on the negative news reported in the news [44]. Exposure to this type of negative political information can subtly and negatively influence the public, potentially leading to negative political attitudes such as political alienation, distrust, and fear. Accordingly, we propose the following hypothesis:

Hypothesis 1: Unofficial media use has a significant deconstruction effect on political trust.

## The mediating role of subjective well-being

Subjective well-being refers to an individual's overall evaluation of their personal quality of life based on their chosen criteria, which includes both cognitive aspects such as life satisfaction and emotional aspects such as positive and negative emotional balance [45, 46]. It is not solely influenced by external factors but also shaped by one's own self-construction, with the assistance of media in the current information age. Due to limitations such as time, space, and cost, people's perception of the objective world is often not solely derived from personal experience but frequently relies on the media. The use of unofficial media platforms, such as social networking sites, enables individuals to stay connected with relatives and friends and promote social interaction, thus contributing to an overall enhancement of their subjective well-being [47, 48]. However, when individuals allocate excessive time to these platforms, they may neglect their daily lives, resulting in a decline in life satisfaction and a weakened sense of subjective well-being [49, 50]. Moreover, negative content on social media is more likely to attract attention than positive content [51], which may further reduce an individual's subjective well-being. Social comparison theory suggests that people often evaluate their own lives by comparing themselves to others. Therefore, if individuals perceive the virtual reality presented on unofficial media as the actual reality, and view the lives of others as more comfortable and superior, their own life satisfaction may decrease [52]. In general, the excessive use of unofficial media platforms, such as Facebook and social networking sites, can undermine subjective well-being [53–57].

Subjective well-being acts as a mediator in the relationship between unofficial media use and political trust. Unofficial media platforms provide a vital outlet for the public to voice their opinions, especially on issues such as corruption, pollution, and food safety. However, such discussions on cyberspace are often characterized by negative information and emotional expressions. A minor mistake can trigger a massive online backlash, which not only diminishes the public's positive emotions but also generates doubt and dissatisfaction towards their lives [58], thus weakening their subjective well-being. Such negative emotions can lead to behaviors such as disengagement, resistance, and even desperate measures to secure the interests of the public. In this process, if national institutions fail to act or respond in a manner that meets the expectations of the public, their needs and interests may not be

fulfilled, further eroding their subjective well-being [59]. This may increase the public's criticism of the current reality and result in a decline in their political trust towards national institutions. Based on attribution theory, the public may attribute this to the lack of managerial or external factors, ultimately diluting their approval of their political institutions. Jin and Nie [60] verify the promoting effect of subjective well-being on the political trust of women and the mediating effect of subjective well-being on the influence of media use on the political trust of women with the help of 2014 Netizens' social consciousness survey data. Accordingly, we propose the following hypothesis:

Hypothesis 2: Subjective well-being plays a mediating role in the influence of unofficial media use on political trust.

## The moderating effects of official media use

This paper not only proposes hypothesis 2 but also delves into another critical issue. It seeks to examine the role of official media in the relationship between subjective well-being and political trust. Official media and unofficial media are inherently relative. If the official media is not treated as a control variable, in the process of unofficial media influencing political trust through the intermediate 'bridge' of subjective well-being, what role and position does the official media play? This is also a question worth exploring. If the mediating effect of subjective well-being exists and the official media has a moderating effect on subjective well-being, which in turn affects political trust, it suggests that there is a complex mechanism of regulating mediating effects in the process of unofficial media influencing political trust. Proposing this complex mechanism hypothesis is meaningful as it helps to comprehensively consider the contingency mechanism of mediating effects and thus more deeply reveal the process of the entire influencing mechanism. The framing effect highlights the media's ability to shape the public's perception of a particular issue [61]. As the 'mouthpiece' of the Party and the government, the official media are strictly supervised by the government and shoulder the mission of propagating political ideas, top-level design and general policies [62]. Over the years, official media has focused on positive publicity, promoting mainstream values, has a positive transformative effect on the public and foster public trust in the government [63]. The unique political attribute of official media has not only won widespread acceptance and persuasion in the process of dissemination but also has performed exceptionally well in improving public awareness and shaping government image [64]. Political trust is an institutional trust that arises from the long-term game and exchange between the public and state institutions [65]. Sufficient information is conducive to the public building a trust relationship with the state institutions in the game and exchange [66]. The research found that political trust is influenced by the government's information disclosure channels, content, and effects [67]. As the preferred channel for state institutions' information disclosure, official media not only improves the transparency of state institutions but also constructs the public's political trust [68]. Therefore, in the path of subjective well-being influencing political trust, official media use can promote the positive impact of subjective well-being on political trust, that is, there is a moderating mediation effect in the process of unofficial media use to deconstruct political trust. we posit the hypothesis that official media use moderates the relationship between subjective well-being and political trust. The strength of official media usage can affect the effect of subjective well-being on political trust. On the one hand, the stronger the official media use, the stronger the positive impact of official media's influence on shaping a positive and optimistic attitude among the public, making them hopeful about life, and thus strengthening the effect of subjective well-being. On the other hand, the weaker the official media use, the more limited the positive impact of

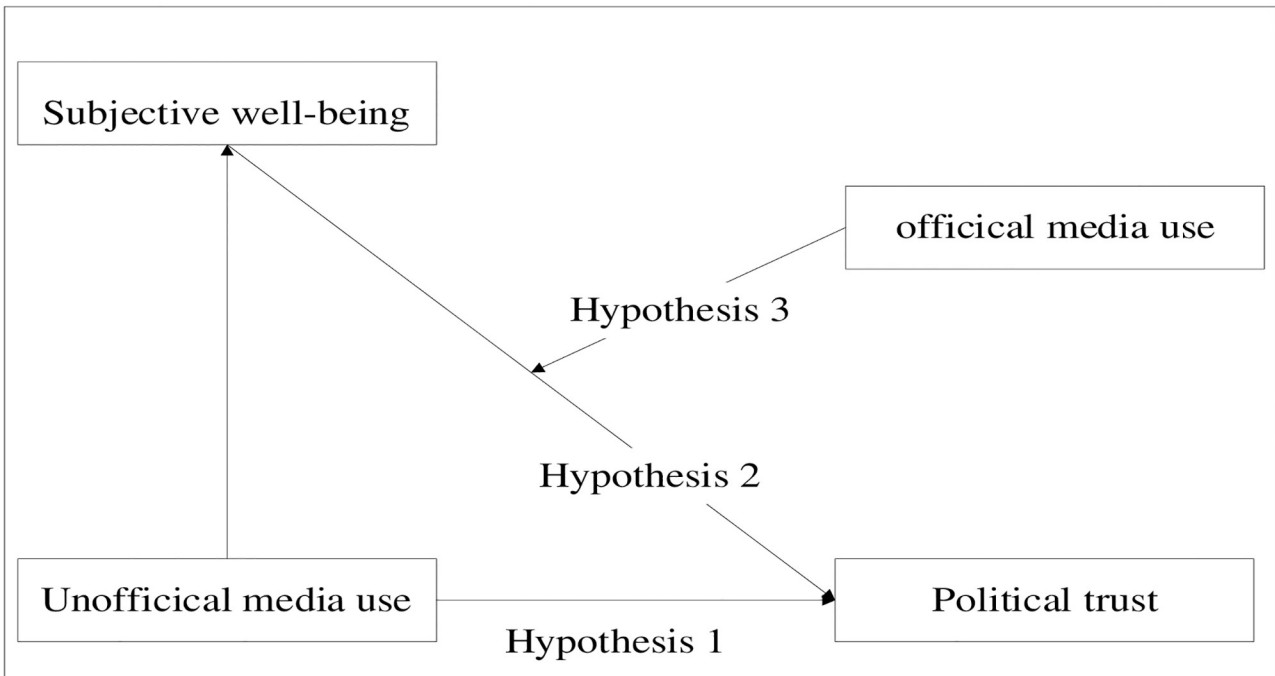

**Fig 1. Theoretical model.**

official media's influence, making the public attribute the decrease in subjective well-being to the failure of government institutions, thereby reducing political trust. Accordingly, we propose the following hypothesis:

Hypothesis 3: Official media use has a positive effect of moderating the relationship between subjective well-being and political trust.

The theoretical model used in the study is illustrated in Fig 1.

## Materials and methods

### Date

The data for this article comes from the 2015 survey on netizens' social consciousness organized and conducted by Professor Ma D. Y. at Renmin University of China. This data is part of the Chinese General Social Survey (CGSS) sub-database, which is a nationally representative, comprehensive, continuous, and representative dataset. Data is available at: http://cgss.ruc.edu.cn/. The survey was conducted using online questionnaires, covering topics such as ideology, social trust, political trust, media exposure, and attitudes towards public opinion issues. The initial sample was 3781, to ensure the reliability of the questionnaire, each IP address was limited to one response, questionnaires completed in less than 8 minutes were excluded, obviously insincere questionnaires were manually reviewed and excluded, and self-evaluation questions were used to assess the authenticity of respondents' answers. After cleaning the data by excluding questionnaires from Hong Kong, Macao, Taiwan, and overseas, respondents under the age of 18, and samples with missing values or those who selected 'I don't know/I don't want to say', a total of 3335 valid samples were obtained.

**Table 1. Descriptive statistics of netizens' trust in various political institutions.**

| Trust at all levels of government | Very distrust = 1 | | Not very much Trust = 2 | | Comparative trust = 3 | | Very trust = 4 | |
|---|---|---|---|---|---|---|---|---|
| | Frequency | (%) | Frequency | (%) | Frequency | (%) | Frequency | (%) |
| Central government trust | 673 | 20.18 | 883 | 26.48 | 1242 | 37.24 | 537 | 16.10 |
| Court trust | 771 | 23.12 | 1186 | 35.56 | 1157 | 34.69 | 221 | 6.63 |
| Police trust | 787 | 23.60 | 1228 | 36.82 | 1164 | 34.90 | 156 | 4.68 |
| Township government trust | 1497 | 44.89 | 1197 | 35.89 | 573 | 17.18 | 68 | 2.04 |

## Measure

Dependent variable. The study's dependent variable is political trust, which refers to the broad political trust and is measured by the index after the item factor analysis of the political trust items. Conventional measurement of political trust mainly includes institutional trust measurement and trust measurement of political institutions and personnel. Referring to the study of Wang and Jin [64], We measure it through the question: 'How much do you trust the central government/court/police/township government?' Responses rang from 1 (very distrust) to 4 (very trusted).

It can be seen from Table 1 that the trust of netizens on various political institutions in descending order is: central government, court, police, and township government. This confirms that political trust is characterized by 'strong central government and weak local government', verifying Li's research [69–71]. The study further reveals that the public tends to trust 'abstract government' more than 'concrete government'. Specifically, the proportion of netizens who report 'very trusted' for the central government is 2.43 times, 3.44 times, and 7.90 times higher than that for the courts, police, and township governments, respectively. Furthermore, the study finds that netizens exhibit the least difference in trust between the courts and police, but the most significant difference in trust between the central government and township government.

Principal component factor analysis was performed on the trust item of each political institution to obtain a common factor named 'political trust'. The rotated factor analysis matrix is shown in Table 2.

Independent variable. The independent variable of the study is unofficial media use, which is measured using an index derived from factor analysis of various questions related to media use. The questionnaire asked participants if they frequently receive political news and current affairs commentary from different channels, including CCTV, Xinhua News Agency, Sina, and other media outlets, with response options ranging from 1 (almost never) to 5 (more than one hour per day). Results from principal component factor analysis revealed two common factors, named 'unofficial media use' and 'official media use', which is consistent with previous

**Table 2. Factor analysis of political trust.**

| Extraction factors and the corresponding original question item | Factor load | Mean | SD |
|---|---|---|---|
| Extraction method is the main component analysis; Maximum variance rotation method; KMO = 0.833; Bartlett test Sig = 0.000; The cumulative variance contribution rate is 74.31% | Political trust | | |
| Central government trust | 0.851 | 2.493 | 0.988 |
| Township government trust | 0.838 | 1.764 | 0.804 |
| Court trust | 0.895 | 2.248 | 0.884 |
| Police trust | 0.863 | 2.207 | 0.854 |

**Table 3. Factor analysis of media use.**

| Extraction factors and the corresponding original question item | Factor load | | Mean | SD |
|---|---|---|---|---|
| Extraction method is the main component analysis; Maximum variance rotation method; KMO = 0.670; Bartlett test Sig = 0.000;The cumulative variance contribution rate is 54.47%. | Unofficial media use | Official media use | | |
| CCTV | -0.033 | 0.889 | 2.184 | 1.043 |
| Xinhua News Agency, People's Daily | 0.098 | 0.871 | 1.864 | 0.923 |
| Sina and other websites | 0.375 | 0.502 | 2.652 | 1.006 |
| Weibo or an online community | 0.672 | -0.187 | 3.203 | 1.041 |
| Wechat | 0.655 | 0.220 | 2.560 | 1.092 |
| Grapevine or chatting with friends | 0.559 | 0.409 | 2.084 | 0.815 |
| Overseas media | 0.645 | 0.012 | 2.192 | 0.962 |

research by Wang and Jin [63] and Chen et al. [42]. Unofficial media use includes platforms such as Weibo or online communities, WeChat, grapevine or chatting with friends, and overseas media. As shown in Table 3, unofficial media has a 'crowding effect' on official media, with netizens showing a preference for unofficial media over official media. Weibo or online communities were found to be the most frequently used, while Xinhua News Agency and People's Daily were the least frequently used official media sources.

Intermediary variable. The study's intermediate variable is subjective well-being, which is measured using a question that is based on the approach used by Fu [72] and Venetoklis [73]. Participants are asked to rate their current level of happiness using a question that reads 'Generally speaking, do you think you are happy now?' with response options ranging from 1 (very unhappy) to 5 (very happy).

Moderating variable. In addition, the study aims to examine the mediating effect of official media use, which is measured using an index derived from factor analysis of questions related to media use. As shown in Table 3, official media use includes CCTV, Xinhua News Agency, People's Daily, and Sina, based on the results of the factor analysis.

Control variables. The study includes several control variables to account for potential confounding factors, such as gender, age, political affiliation, annual household income, residence, educational background, political discussion, frequency of netizens' exposure to news, and regional-fixed effect. The detailed description of each variable is presented in Table 4.

## Model selection

Bootstrap method. Common tests for mediating effects include stepwise regression, product of coefficients, coefficient of variation and Bootstrap, which is currently the preferred method for testing mediating effects because of its high statistical validity compared to other tests for mediating effects [74]. In contrast to repeated sampling methods, other statistical tests (such as the Sobel test) can only be used to test specific composite coefficients [75]. Therefore, the Bootstrap method was primarily chosen for this paper to test the relationship between unofficial media use and political trust. The Bootstrap method works by generating a number of re-samples from the original dataset and making replacements. This means that each re-sample is obtained by randomly selecting observations from the original dataset and adding them to the re-sample.

The propensity value matches. Netizens' behaviour in using unofficial media is not random, but can be influenced by a number of factors. This suggests that there may be a self-selection problem in the use of unofficial media. Propensity value matching is an effective method to

**Table 4. Descriptive statistical analysis of each variable.**

| Variable | Description of the problem | Mean | SD | Min | Max |
|---|---|---|---|---|---|
| Politics trust | Central government trust | 2.493 | 0.988 | 1 | 4 |
| | Court trust | 2.248 | 0.884 | 1 | 4 |
| | Police trust | 2.207 | 0.854 | 1 | 4 |
| | Township government trust | 1.764 | 0.804 | 1 | 4 |
| | Political trust factor | 7.515 | 2.625 | 3.447 | 13.786 |
| Unofficial media use | Weibo or online communities | 3.203 | 1.041 | 1 | 5 |
| | WeChat | 2.560 | 1.092 | 1 | 5 |
| | Grapevine or chatting with friends | 2.084 | 0.815 | 1 | 5 |
| | Overseas media | 2.192 | 0.962 | 1 | 5 |
| | Unofficial media use factor | 6.409 | 1.634 | 2.532 | 12.658 |
| Subjective well-being | Generally speaking, do you think you are happy now | 3.228 | 0.957 | 1 | 5 |
| Official media use | CCTV | 2.184 | 1.043 | 1 | 5 |
| | Xinhua News Agency, People's Daily | 1.864 | 0.923 | 1 | 5 |
| | Sina and other websites | 2.652 | 1.006 | 1 | 5 |
| | Official media use factor | 4.900 | 1.839 | 2.262 | 11.309 |
| Gender | Female = 0; Male = 1 | 0.704 | 0.457 | 0 | 1 |
| Age | Continuous variable | 34.161 | 10.451 | 18 | 83 |
| Age$^2$/100 | The square of the age was divided by 100 | 12.762 | 7.974 | 3.240 | 68.890 |
| Political affiliation | Non-party member = 0; Party member = 1 | 0.261 | 0.440 | 0 | 1 |
| Annual household income | Under $100,000 = 1; $100,000-$500,000 = 2; More than 500,000 = 3 | 1.551 | 0.599 | 1 | 3 |
| Residence | Village = 0; City = 1 | 0.970 | 0.171 | 0 | 1 |
| Education background | Under undergraduate course = 1; Undergraduate course = 2; Bachelor or above degree = 3 | 1.908 | 0.685 | 1 | 3 |
| Political discussion | Do you often talk to others about international and domestic political, economic and social issues? | 1.792 | 0.643 | 1 | 4 |
| Frequency of exposure to the news | Do you often tune in, listen to, or read news or current events on television, radio, newspaper, or website? | 1.579 | 0.783 | 1 | 4 |
| Region | Eastern = 1; Middle part = 2; Westward = 3 (dummy variables) | 1.471 | 0.730 | 1 | 3 |

address the selectivity bias. The method constructs a counterfactual group (control group) based on a counterfactual framework, and then estimates the difference in the dependent variable between the experimental and control groups, all else being equal (i.e. the mean intervention effect ATT), and thus infers the direction and extent of the effect of the intervening variable on the dependent variable. However, while traditional propensity value matching is mainly used to analyse dichotomous intervention variable scenarios, political trust is a continuous variable in this paper, and a more scientific generalised propensity value matching method is required [76].

## Empirical results

### Main-effect test

The study employs the bootstrap method to verify the hypotheses. In Table 5, column (1) shows the regression of unofficial media use on political trust only, and the results show that the regression coefficient is significantly negative at the 5% statistical level. In column (2), we introduce additional control variables, and the coefficient of unofficial media use on political trust increases by 18.5%. This finding supports the notion that unofficial media use has a negative impact on political trust, which confirms the media depression theory. Thus, hypothesis 1 is supported.

**Table 5. Main effect, mediating effect, and moderated mediating effect.**

| | Main effect | | Mediating effect | | Moderated mediating effect | | | |
| --- | --- | --- | --- | --- | --- | --- | --- | --- |
| | (1) | (2) | (3) | (4) | (5) | (6) | (7) | (8) |
| | Political trust | Political trust | Subjective well-being | Political trust | Political trust | Subjective well-being | Political trust | Political trust |
| Unofficial media use | -0.102** | -0.098** | -0.026* | -0.067* | -0.254*** | -0.060*** | -0.205*** | -0.220*** |
| | (0.032) | (0.030) | (0.011) | (0.027) | (0.024) | (0.010) | (0.022) | (0.023) |
| Subjective well-being | | | | 1.181*** | | | 0.820*** | 0.433*** |
| | | | | (0.041) | | | (0.040) | (0.095) |
| Official media use | | | | | 0.760*** | 0.162*** | 0.627*** | 0.343*** |
| | | | | | (0.021) | (0.009) | (0.022) | (0.073) |
| Official media use×Subjective well-being | | | | | | | | 0.083*** |
| | | | | | | | | (0.020) |
| Gender | | -1.230*** | -0.404*** | -0.753*** | -0.936*** | -0.341*** | -0.656*** | -0.656*** |
| | | (0.094) | (0.034) | (0.088) | (0.078) | (0.033) | (0.077) | (0.076) |
| Age | | -0.093*** | -0.027** | -0.061** | -0.105*** | -0.030** | -0.081*** | -0.086*** |
| | | (0.023) | (0.009) | (0.020) | (0.020) | (0.009) | (0.018) | (0.018) |
| Age$^2$/100 | | 0.021 | 0.015 | 0.003 | 0.065* | * | 0.045 | 0.052* |
| | | (0.029) | (0.012) | (0.026) | (0.026) | (0.012) | (0.023) | (0.023) |
| Political affiliation | | 0.840*** | 0.215*** | 0.586*** | 0.461*** | 0.135*** | 0.351*** | 0.344*** |
| | | (0.099) | (0.038) | (0.089) | (0.083) | (0.036) | (0.078) | (0.076) |
| Annual household income | | 0.095 | 0.313*** | -0.275*** | -0.034 | 0.286*** | -0.268*** | -0.263*** |
| | | (0.087) | (0.035) | (0.080) | (0.0768) | (0.034) | (0.072) | (0.072) |
| Residence | | 0.494* | 0.260* | 0.187 | 0.328 | 0.224* | 0.144 | 0.157 |
| | | (0.242) | (0.108) | (0.226) | (0.232) | (0.102) | (0.221) | (0.223) |
| Education background | | -0.202** | 0.062** | -0.275*** | -0.072 | 0.090*** | -0.146** | -0.135** |
| | | (0.064) | (0.024) | (0.058) | (0.053) | (0.023) | (0.051) | (0.049) |
| Political discussion | | -0.058 | -0.095*** | 0.054 | 0.153* | -0.050 | 0.194*** | 0.217*** |
| | | (0.073) | (0.028) | (0.067) | (0.0614) | (0.027) | (0.058) | (0.057) |
| Frequency of exposure to the news | | -0.194*** | -0.084 | -0.093*** | 0.149** | -0.020 | 0.165*** | 0.150*** |
| | | (0.053) | (0.023) | (0.048) | (0.047) | (0.022) | (0.043) | (0.043) |
| Constant | 8.170*** | 11.590*** | 3.629*** | 7.307*** | 7.911*** | 2.844*** | 5.580*** | 6.987*** |
| | (0.198) | (0.545) | (0.217) | (0.539) | (0.5042) | (0.214) | (0.484) | (0.586) |
| Region | No | Yes | Yes | Yes | Yes | Yes | Yes | Yes |
| R$^2$ | 0.004 | 0.189 | 0.133 | 0.350 | 0.426 | 0.214 | 0.497 | 0.499 |
| N | 3335 | 3335 | 3335 | 3335 | 3335 | 3335 | 3335 | 3335 |

Note. (1) Boot standard errors are reported in parentheses; (2) In the Bootstrap method, 5000 times of repeated sampling are used; (3) In the analysis of the moderating effect, both subjective well-being and the use of official media are centralized; (4)

*p<0.10;

**p<0.05;

***p<0.01.

## Test of mediating effect of subjective well-being

Table 5, column (3) presents the estimated coefficient of unofficial media use on subjective well-being is significantly negative at the 10% level of significance. Furthermore, in column (4), unofficial media use is negatively associated with political trust at the 10% level of significance, subjective well-being is positively associated with political trust at the 1% level of significance. These findings provide evidence supporting the mediating role of subjective well-being

in the relationship between unofficial media use and political trust, thereby confirming hypothesis 2.

## Test of moderated mediating effect of official media use

Column (5) in Table 5 demonstrates a significant negative impact of unofficial media use on political trust at the 1% statistical level, based on the results from the regression analysis using column (2) as the baseline. In column (6), official media use is added to the model from column (3), and the estimated coefficient of unofficial media use on subjective well-being is found to be significant at the 1% level of significance. In column (7), both unofficial media use and subjective well-being are found to have a significant effect on political trust at the 1% statistical level. Finally, in column (8), the interaction term of official media use and subjective well-being is added, and the estimated coefficient of the interaction term on political trust is positive at the 1% significance level, indicating that the use of official media has a positive moderating effect on subjective well-being. Thus, hypothesis 3 is confirmed.

To provide a more visual representation of the moderating effect of official media use, the study uses a mediated moderation approach to illustrate the relationship between subjective well-being and political trust. The simple slope method is employed to divide official media use into high and low categories based on the mean plus or minus one standard deviation. Fig 2 presents an adjustment schematic diagram, which shows that as the use of official media increases, the predictive effect of subjective well-being on political trust gradually strengthens. Specifically, the positive predictive effect of subjective well-being on political trust is stronger for netizens with high use of official media compared to those with low use of official media.

## The propensity value matches

To address the issue of selective bias in the study, the propensity value matching method is employed, specifically the generalized propensity score matching method (GPS), since the variable for unofficial media use is continuous. The first step involves estimating the conditional probability distributions and generalized propensity scores for the use of unofficial media. In the second step, a balance test is conducted to ensure the comparability of the treatment and control groups, and the GPS values obtained in the first step are used as control variables in the ordinary least squares estimation. Finally, the study estimates the response function by

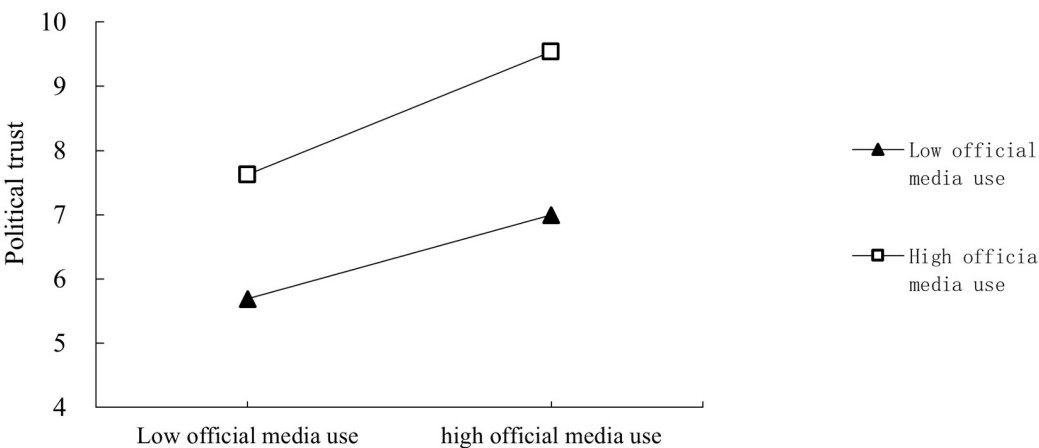

**Fig 2. Moderating effects of official media use.**

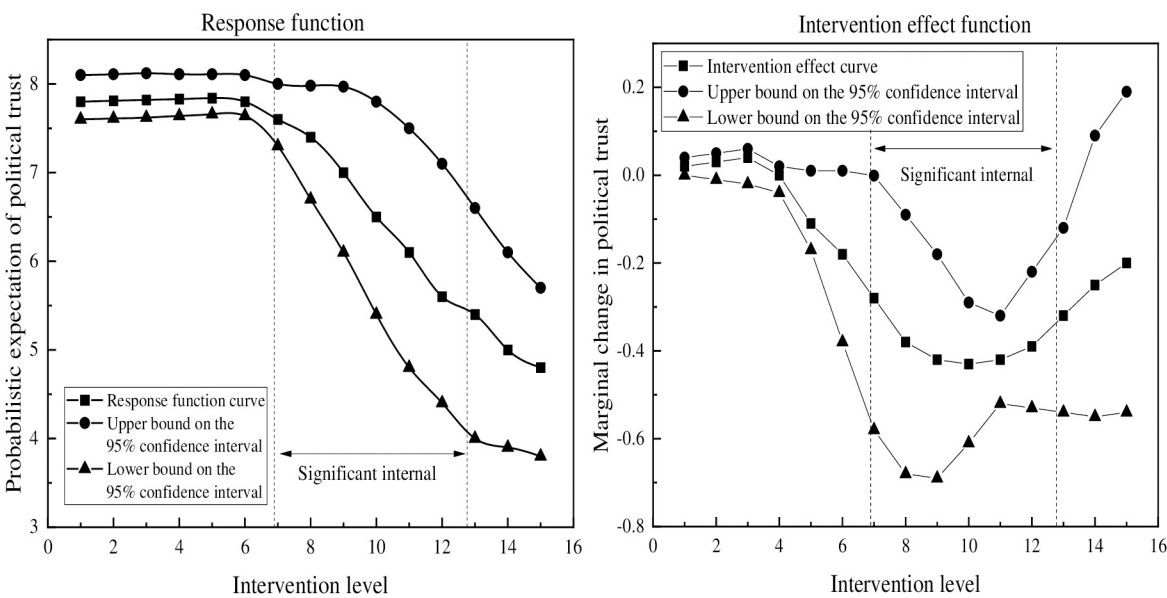

**Fig 3. Main effect of unofficial media on political trust.**

dividing the range of unofficial media use [2.532,12.658] into five sub-intervals [2.532,4.436], [4.465,6.391], [6.397,8.448], [8.454,10.798], [10.809,12.658] to examine the causal impact of unofficial media use on political trust in each interval.

Fig 3 illustrates the intervention effect of unofficial media use on political trust. On the left-hand side of the figure, the expected probability of political trust is shown across different levels of intervention. The results indicate that when the index of unofficial media use ranges from 2.5 to 5, the expected probability of political trust increases from 7.812 to 7.840. However, when the index increases from 5 to 12.7, the expected probability of political trust decreases from 7.840 to 5.463. The right-hand side of the figure displays the marginal impact of unofficial media use on political trust, revealing a statistically significant decrease followed by an increase in political trust, with the negative impact occurring when the index of unofficial media use exceeds 6.9. This suggests that unofficial media use has a substantial negative impact on political trust when the index surpasses 6.9.

Fig 4 shows the intervention effect of unofficial media use on political trust in the mediating effect. As the index of unofficial media use increases, political trust initially shows a gradual improvement, but eventually experiences a sharp decline. The intervention effect function suggests that the usage index of unofficial media range of 6.9 to 12.7 is where the impact of unofficial media on political trust is most significant. To summarize, the unofficial media use has a non-linear relationship with political trust, with an optimum usage range that elicits the most significant effect.

Based on Fig 5, the moderated mediating effect indicates a declining trend in the influence of unofficial media use on political trust. The graph suggests that the impact of unofficial media use on political trust decreases as time goes on. Upon further examination of the intervention effect function, it is evident that a significant negative correlation between unofficial media use and political trust emerges when the usage index of unofficial media falls within the range of 2.5 to 7.9. In other words, within this specific range, an increase in the use of unofficial media corresponds with a decrease in political trust.

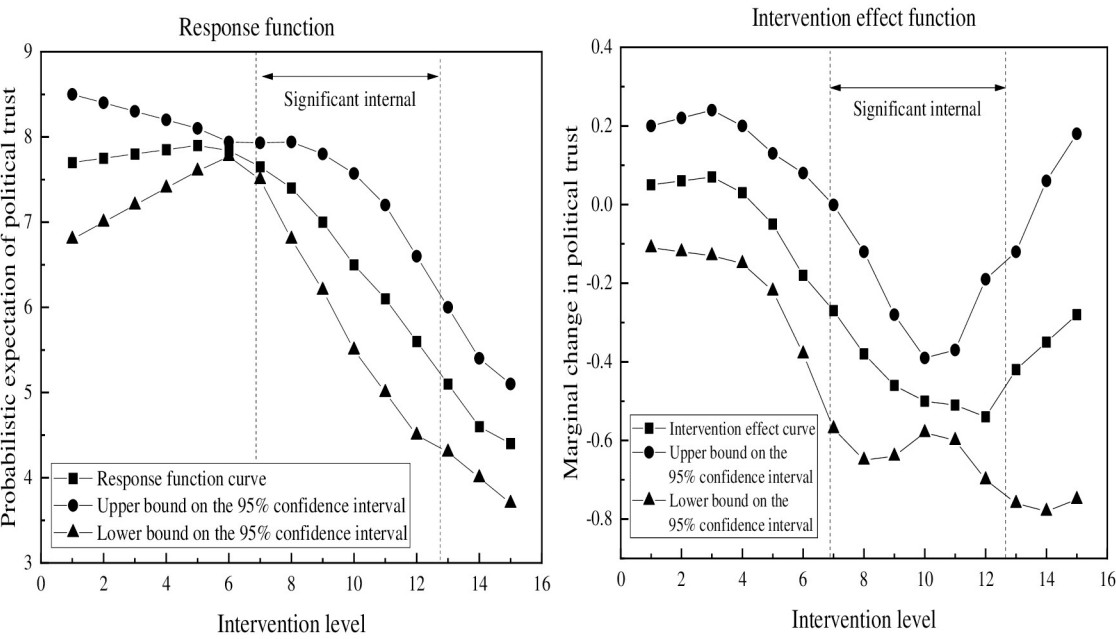

**Fig 4. Mediating effect of unofficial media on political trust.**

After analyzing Figs 3 through 5 and Table 6, it is evident that there exists a non-linear relationship between the use of unofficial media and political trust. As the use of unofficial media increases, the probability of political trust decreases. However, the range of this significant effect varies depending on the conditions of the analysis. Specifically, as the analysis progresses from main effect to mediating effect and then to moderated mediating effect, the significant

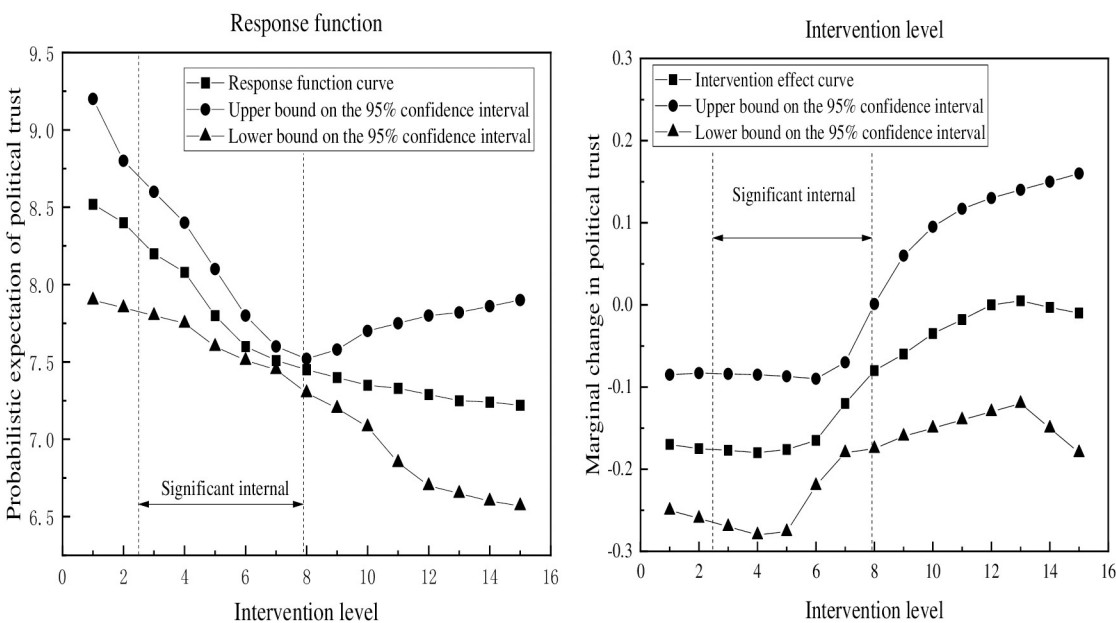

**Fig 5. Moderated mediating effect of unofficial media use on political trust.**

**Table 6. Summary of significant intervals for each effect.**

| Effect | Section | Political trust | |
|---|---|---|---|
| | | Take value change | Chang trend |
| Main effect | (6.9,12.7) | 7.631→5.463 | Downward |
| Mediating effect | (6.9,12.7) | 7.662→5.175 | Downward |
| Moderated mediating effect | (2.5,7.9) | 8.355→7.446 | Downward |

deconstruction range of political trust in the unofficial media use becomes narrower, shifting from (6.9,12.7) to (6.9,7.9) public range.

## Placebo test

A placebo test was carried out to investigate the specific influence of unofficial media use on political trust, while controlling for other factors associated with netizens. The objective was to determine whether there are inherent factors in netizens that lead to unofficial media use and subsequent erosion of political trust. To achieve this, individual netizens were randomly assigned to use unofficial media while keeping other control variables and individual characteristics constant. The results of the placebo test, as observed in Table 7, indicate that unofficial media use does not have a causal relationship with political trust independent of other netizens' characteristics. This absence of any significant effect of the use of unofficial media on political trust, from the main effect to the mediating effect and ultimately the moderated mediating effect, suggests that the use of unofficial media does not have a direct impact on political trust. The placebo test results help to rule out any

**Table 7. Placebo test results.**

| | Main effect | | Mediating effect | | Moderated mediating effect | | |
|---|---|---|---|---|---|---|---|
| | (1) | (2) | (3) | (4) | (5) | (6) | (7) |
| | Political trust | Subjective well-being | Political trust | Political trust | Subjective well-being | Political trust | Political trust |
| Placebo metrics | -0.008 | -0.005 | -0.002 | -0.011 | -0.005 | -0.007 | -0.007 |
| | (0.025) | (0.009) | (0.023) | (0.022) | (0.009) | (0.020) | (0.020) |
| Subjective well-being | | | 1.196*** | | | 0.865*** | 0.615*** |
| | | | (0.041) | | | (0.038) | (0.096) |
| Official media use | | | | 0.719*** | 0.153*** | 0.587*** | 0.399*** |
| | | | | (0.021) | (0.009) | (0.020) | (0.069) |
| Official media use×Subjective well-being | | | | | | | 0.054** |
| | | | | | | | (0.019) |
| Constant | 10.940*** | 3.491*** | 6.762*** | 6.340*** | 2.513*** | 4.165*** | 5.020*** |
| | (0.548) | (0.206) | (0.510) | (0.486) | (0.204) | (0.463) | (0.552) |
| Control variable | Yes | Yes | Yes | Yes | Yes | Yes | Yes |
| $R^2$ | 0.1860 | 0.1340 | 0.3508 | 0.4056 | 0.2086 | 0.4844 | 0.4856 |
| N | 3335 | 3335 | 3335 | 3335 | 3335 | 3335 | 3335 |

Note. (1) Standard errors are reported in parentheses; (2) In the analysis of the moderating effect, both subjective well-being and the use of official media are centralized; (3)

*$p<0.10$;

**$p<0.05$;

***$p<0.01$

confounding factors that may be associated with the use of unofficial media, providing further evidence to support the conclusion that unofficial media use does not have a significant impact on political trust.

## Heterogeneity analysis

Table 5 presents a regression analysis that supports the notion that unofficial media use can negatively impact political trust. However, it is still unclear whether this effect varies across different political institutions. To address this question, we conducted separate multiple ordered logistic regression analyses for trust in four specific institutions: the central government, court, police, and township government. The findings from each model are presented in Tables 8 and 9. By examining these results, we can gain a deeper understanding of how unofficial media use affects trust in distinct political institutions.

Table 8 presents the results of our analysis on the impact of unofficial media use on trust in specific political institutions. Our findings indicate that unofficial media use can significantly reduce trust in the central government, court, and police. Furthermore, we observe that this effect is mediated by a decrease in subjective well-being. However, we do not find any significant association between unofficial media use and trust in township governments. Overall, our results suggest that the impact of unofficial media use on political trust varies across different institutions, with some being more susceptible to deconstruction than others.

Our analysis in Table 9 reveals that official media use can have a positive moderating effect on the relationship between subjective well-being and trust in specific political institutions. Specifically, we observe that the mediating effect of subjective well-being on trust in the central government, court, and police is strengthened by official media use. These findings highlight the potential role of official media use in mitigating the negative impact of subjective well-being on political trust.

The regression analyses presented in Tables 8 and 9 demonstrate that the impact of unofficial media use on political trust varies depending on the type of political institution in question. This raises the question of whether different types of unofficial media have differing

**Table 8. Mediating effect: Multiple ordered logistic regression results of trust in different political institutions.**

| | (1) | (2) | (3) | (4) | (5) | (6) | (7) | (8) | (9) |
|---|---|---|---|---|---|---|---|---|---|
| | Central government trust | Township government trust | Court trust | Police trust | Subjective well-being | Central government trust | Township government trust | Court trust | Police trust |
| Unofficial media use | 0.904*** | 0.978 | 0.913*** | 0.904*** | 0.948* | 0.923*** | 1.000 | 0.936** | 0.927** |
| | (0.024) | (0.026) | (0.025) | (0.025) | (0.025) | (0.023) | (0.025) | (0.025) | (0.025) |
| Subjective well-being | | | | | | 2.633*** | 2.508*** | 2.310*** | 2.317*** |
| | | | | | | (0.042) | (0.048) | (0.041) | (0.043) |
| Control variable | Yes | Yes | Yes | Yes | Yes | Yes | Yes | Yes | Yes |
| Pseudo $R^2$ | 0.055 | 0.071 | 0.079 | 0.056 | 0.057 | 0.124 | 0.136 | 0.134 | 0.113 |
| Log likelihood | -4212.68 | -3438.61 | -3849.77 | -3840.63 | -4181.57 | -3904.07 | -3197.47 | -3618.11 | -3607.43 |
| N | 3335 | 3335 | 3335 | 3335 | 3335 | 3335 | 3335 | 3335 | 3335 |

Note. (1) In multiple ordered logistic regression, the value is the occurrence ratio; (2) In the analysis of the moderating effect, both subjective well-being and the use of official media are centralized; (3)

*p<0.10;

**p<0.05;

***p<0.01.

**Table 9. Moderated mediating effect: Multiple ordered logistic regression results of trust in different political institutions.**

| | (1) | (2) | (3) | (4) | (5) | (6) | (7) | (8) | (9) | (10) |
|---|---|---|---|---|---|---|---|---|---|---|
| | Central government trust | Court trust | Police trust | Subjective well-being | Central government trust | Court trust | Police trust | Central government trust | Court trust | Police trust |
| Unofficial media use | 0.773*** | 0.819*** | 0.821*** | 0.881*** | 0.796*** | 0.845*** | 0.848*** | 0.790*** | 0.837*** | 0.837*** |
| | (0.024) | (0.025) | (0.025) | (0.024) | (0.024) | (0.026) | (0.025) | (0.025) | (0.025) | (0.025) |
| Subjective well-being | | | | | 2.089*** | 1.880*** | 1.933*** | 1.558*** | 1.388** | 1.285* |
| | | | | | (0.046) | (0.044) | (0.043) | (0.119) | (0.116) | (0.112) |
| Official media use | 2.057*** | 1.742*** | 1.650*** | 1.465*** | 1.902*** | 1.607*** | 1.521*** | 1.535*** | 1.286** | 1.126 |
| | (0.024) | (0.024) | (0.023) | (0.022) | (0.025) | (0.025) | (0.023) | (0.087) | (0.084) | (0.084) |
| Official media use×Subjective well-being | | | | | | | | 1.066* | 1.067** | 1.093*** |
| | | | | | | | | (0.025) | (0.024) | (0.024) |
| Control variable | Yes | Yes | Yes | Yes | Yes | Yes | Yes | Yes | Yes | Yes |
| Pseudo R$^2$ | 0.181 | 0.162 | 0.127 | 0.097 | 0.216 | 0.190 | 0.158 | 0.217 | 0.191 | 0.161 |
| Log likelihood | -3652.54 | -3504.80 | -3551.83 | -4003.52 | -3495.06 | -3387.49 | -3422.08 | -3491.03 | -3382.77 | -3413.35 |
| N | 3335 | 3335 | 3335 | 3335 | 3335 | 3335 | 3335 | 3335 | 3335 | 3335 |

Note. (1) In multiple ordered logistic regression, the value is the occurrence ratio; (2) In the analysis of the moderating effect, both subjective well-being and the use of official media are centralized; (3) Since subjective well-being has no mediating effect on the trust of township governments through the use of unofficial media, Table 9 does not report the mediating model that moderates the trust of township governments. (4)

*p<0.10;

**p<0.05;

***p<0.01.

effects on political trust. To explore this, multiple regression models were constructed and presented in Table 10. The results from columns (1) to (4) indicate that the use of Weibo or online communities, as well as overseas media, can have a direct negative impact on political trust. Alternatively, they may indirectly deconstruct political trust through subjective well-being. On the other hand, chatting with friends or using grapevine media can have a direct positive impact on political trust, or indirectly construct political trust through subjective well-being. In columns (5) to (8), it is shown that while CCTV has a moderating effect on the relationship between subjective well-being and political trust, the Xinhua News Agency, People's Daily, and Sina website do not exhibit this moderating effect. This suggests that the impact of different types of unofficial media on political trust can vary significantly, and the nature of this impact may be moderated by certain factors.

## Discussion

Research has shown that the media has the power to influence and mold the public's political interests, knowledge, attitudes, emotions, and culture [77–79]. This is particularly true in the Chinese media landscape, which has a distinct set of characteristics that set it apart from Western media [43]. The rapid development of science and technology is also bringing about significant changes in media ecology and structure. As a crucial link between the subject and object of political trust, the media has an important role to play in fostering political trust. However, there is a lack of research on the impact of unofficial media use on political trust. This study aims to address this gap by examining the effects of unofficial media on political trust, as well as the underlying mechanisms and any relevant variations. Meanwhile, the research findings provide theoretical basis and empirical experience for enhancing government trust and

**Table 10. The impact of different types of unofficial official media use on political trust.**

| | Main effect | | Mediating effect | | Moderated mediating effect | | |
| --- | --- | --- | --- | --- | --- | --- | --- |
| | (1) | (2) | (3) | (4) | (5) | (6) | (7) |
| | Political trust | Political trust | Subjective well-being | Political trust | Political trust | Subjective well-being | Political trust |
| Weibo or Online communities | -0.447*** | -0.435*** | -0.091*** | -0.336*** | -0.304*** | -0.066*** | -0.252*** |
| | (0.044) | (0.041) | (0.016) | (0.037) | (0.037) | (0.016) | (0.034) |
| WeChat | 0.043 | 0.105* | 0.013 | 0.090* | -0.013 | -0.013 | -0.003 |
| | (0.045) | (0.041) | (0.017) | (0.037) | (0.034) | (0.017) | (0.032) |
| Grapevine or chatting with friends | 0.797*** | 0.570*** | 0.129*** | 0.429*** | 0.159** | 0.039 | 0.129** |
| | (0.064) | (0.060) | (0.022) | (0.054) | (0.051) | (0.022) | (0.048) |
| Overseas media | -0.494*** | -0.416*** | -0.114*** | -0.292*** | -0.402*** | -0.113*** | -0.314*** |
| | (0.050) | (0.046) | (0.019) | (0.041) | (0.038) | (0.018) | (0.036) |
| Subjective well-being | | | | 1.090*** | | | 0.779*** |
| | | | | (0.041) | | | (0.040) |
| CCTV | | | | | 0.656*** | 0.119*** | 0.563*** |
| | | | | | (0.051) | (0.021) | (0.046) |
| Xinhua News Agency, People's Daily | | | | | 0.712*** | 0.171*** | 0.579*** |
| | | | | | (0.057) | (0.022) | (0.055) |
| Sina and other websites | | | | | 0.172*** | 0.0496** | 0.133*** |
| | | | | | (0.037) | (0.017) | (0.035) |
| Constant | 8.256*** | 11.63*** | 3.737*** | 7.558*** | 8.332*** | 3.030*** | 5.971*** |
| | (0.197) | (0.523) | (0.220) | (0.515) | (0.503) | (0.216) | (0.482) |
| Control variable | No | Yes | Yes | Yes | Yes | Yes | Yes |
| $R^2$ | 0.103 | 0.2552 | 0.1654 | 0.3871 | 0.4489 | 0.2325 | 0.5109 |
| N | 3335 | 3335 | 3335 | 3335 | 3335 | 3335 | 3335 |

| | Moderated mediating effect | | | | | | |
| --- | --- | --- | --- | --- | --- | --- | --- |
| | (8) | (9) | (10) | (11) | (12) | (13) | (14) |
| | Political trust | Political trust | Political trust | Political trust | Political trust | Political trust | Political trust |
| Weibo or Online communities | -0.259*** | -0.239*** | -0.264*** | -0.410*** | -0.231*** | -0.274*** | -0.306*** |
| | (0.033) | (0.034) | (0.034) | (0.037) | (0.033) | (0.035) | (0.035) |
| WeChat | -0.016 | 0.008 | 0.020 | 0.029 | -0.006 | -0.009 | -0.004 |
| | (0.033) | (0.034) | (0.034) | (0.037) | (0.033) | (0.033) | (0.034) |
| Grapevine or chatting with friends | 0.135** | 0.226*** | 0.162** | 0.351*** | 0.143** | 0.208*** | 0.141** |
| | (0.048) | (0.049) | (0.050) | (0.054) | (0.048) | (0.049) | (0.051) |
| Overseas media | -0.334*** | -0.282*** | -0.377*** | -0.292*** | -0.333*** | -0.281*** | -0.371*** |
| | (0.036) | (0.038) | (0.038) | (0.041) | (0.036) | (0.037) | (0.038) |
| Subjective well-being | 0.447*** | 0.482*** | 0.646*** | 0.680*** | 0.481*** | 0.474*** | 0.512*** |
| | (0.100) | (0.080) | (0.081) | (0.097) | (0.084) | (0.100) | (0.104) |
| CCTV | -0.043 | 0.286* | | | -0.008 | 0.222 | |
| | (0.170) | (0.134) | | | (0.169) | (0.143) | |
| CCTV×Subjective well-being | 0.180*** | 0.180*** | | | 0.178*** | 0.184*** | |
| | (0.049) | (0.037) | | | (0.049) | (0.039) | |
| Xinhua News Agency, People's Daily | 0.645** | | 0.597*** | | 0.696*** | | 0.563*** |
| | (0.206) | | (0.154) | | (0.211) | | (0.166) |
| Xinhua News Agency, People's Daily×Subjective well-being | -0.023 | | 0.119** | | -0.032 | | 0.109* |
| | (0.059) | | (0.043) | | (0.060) | | (0.046) |
| Sina and other websites | 0.137 | | | -0.018 | | 0.200 | 0.053 |
| | (0.106) | | | (0.111) | | (0.111) | (0.104) |

(*Continued*)

**Table 10.** (Continued)

| | | | | | | | |
|---|---|---|---|---|---|---|---|
| Sina and other websites ×Subjective well-being | -0.002 | | | 0.138*** | | -0.005 | 0.053 |
| | (0.033) | | | (0.035) | | (0.034) | (0.033) |
| Constant | 7.217*** | 7.507*** | 7.507*** | 8.199*** | 7.311*** | 7.304*** | 7.643*** |
| | (0.584) | (0.576) | (0.553) | (0.603) | (0.569) | (0.601) | (0.580) |
| Control variable | Yes | Yes | Yes | Yes | Yes | Yes | Yes |
| $R^2$ | 0.5147 | 0.4936 | 0.4836 | 0.4120 | 0.5140 | 0.4973 | 0.4893 |
| N | 3335 | 3335 | 3335 | 3335 | 3335 | 3335 | 3335 |

Note. (1) Standard errors are reported in parentheses; (2) In the analysis of the moderating effect, both subjective well-being and the use of official media are centralized;
(3)
*p<0.10;
**p<0.05;
***p<0.01.

constructing modern national governance, and also provide some reference for countries with similar backgrounds to China.

Table 8 indicates that unofficial media use does not significantly affect the trust levels of township governments. We propose three possible explanations for this finding. Firstly, the 'knowledge gap hypothesis' suggests that information dissemination not only widens the information acquisition gap between different social groups [80] but also exacerbates differences in their knowledge, attitudes, and behaviors [81]. Despite the growing number of Chinese netizens each year, there is still a notable disparity in internet literacy between those residing in urban and rural areas. This is due to factors such as varying levels of economic development, disparities in education resources and infrastructure facilities. As a result, people living in villages and towns have limited access to unofficial media channels, which in turn limits their ability to process and screen information effectively, and leads to discrepancies in terms of information acquisition, knowledge acquisition, political attitudes, and political behaviors, as compared to their urban counterparts. Secondly, as the ultimate implementers of central government policies, township governments interact more frequently with netizens, which may lead to netizens' trust and evaluation of township governments relying more on their direct interaction experiences in real life. Thirdly, Resonating with the audience is one of the main objectives of media coverage, and non-official media is no exception. To maximize benefits, the 'gatekeepers' of unofficial media selectively report some political information based on the breadth, depth, and influence of the content in the sea of political information. This selective reporting may result in a lower frequency of coverage of township government information compared to other political institutions, making it difficult to form a scale effect and limiting the spread of township government information. Furthermore, information dissemination exhibits a certain proximity diffusion phenomenon, where the effectiveness of information dissemination weakens as the geographical distance increases [82], resulting in low attention and resonance among netizens for township government information.

Table 10 demonstrates that engaging in grapevine discussions or chatting with friends can actually bolster political trust, contrary to popular belief that such discussions may lead to a deconstruction effect. This study presents two potential reasons for this finding. In terms of survey questions, social trust in China is mainly built on personal networks and the public prefers and trusts information obtained through acquaintances. When it comes to obtaining political information through gossip or chatting with friends, the majority of netizens may obtain more information from chatting with friends. The political information obtained from

chatting with friends is not all negative, which may lead to the 'concealment' of the deconstructive effect of gossip on political trust, resulting in a 'distorted' effect of gossip on political trust. In terms of field and context, Weibo or online communities are relatively open, and information is mainly obtained through public channels. Netizens have lower trust in it, and positive information has far less impact on netizens than negative information, which increases the difficulty of building political trust. On the other hand, rumors are unconfirmed and undisclosed information, often with a mysterious color, which can easily arouse netizens' curiosity [83]. Moreover, rumors always occur in closed fields, are often focal information, obtained through private channels, and closely related to the interests of the parties involved, making it easier to resonate and build political trust among netizens.

It is noteworthy that Xinhua News Agency, People's Daily, and Sina website do not appear to moderate the relationship between subjective well-being and political trust. There are a few possible explanations for this finding. Firstly, Xinhua News Agency and People's Daily may not have the same level of vividness as CCTV, as suggested by the lower usage frequency of netizens for Xinhua News Agency and People's Daily compared to CCTV. As a result, their role as intermediaries may not have as much of an impact on this relationship. Secondly, in contrast to CCTV, Sina websites not only serve as a means of information transmission but also provide a platform for free online expression. However, during interactions with users, negative comments tend to garner more attention than positive ones, potentially resulting in a 'covering up' of positive effects by the negative effects of reviews. This phenomenon can impact the output of information on Sina websites. Thirdly, It is possible that political information reported by CCTV, Xinhua News Agency, People's Daily, Sina, and other websites may contain repetitive content. This repetition could result in the positive moderating effect of CCTV 'covering up' the moderating effects of other websites such as Xinhua News Agency, People's Daily, and Sina. To test this theory, three moderating variables are added to columns (9) through (11) of Table 10. The results demonstrate that CCTV, Xinhua News Agency, People's Daily, and Sina had a positive moderating effect. In columns (12) to (14), the three moderating variables are combined in pairs. The results reveal that when CCTV was combined with the other two moderating variables, its moderating effects 'masked' those of the other variables. Similarly, when Xinhua News Agency, People's Daily, Sina, and other websites are combined, the moderating effect of Sina and other websites is 'masked' by the moderating effect of Xinhua News Agency and People's Daily. This finding supports the third conjecture.

## Conclusion

This study mainly uses the 2015 Chinese netizens' survey on social consciousness to analyze the impact of unofficial media use on political trust. The research results support our hypotheses. In terms of the main effects, unofficial media use significantly deconstructs political trust. In terms of mechanisms, first, subjective well-being plays a mediating role in the impact of unofficial media use on political trust. Second, official media use has a moderating effect on the relationship between subjective well-being and political trust. In terms of heterogeneity, unofficial media use has different effects on trust in different political institutions. Specifically, the deconstructive effect of unofficial media use on trust in the central government, court, and police is stronger than that on trust in local government. Different types of unofficial media use also have different effects on political trust. Specifically, microblogs, online communities, and overseas media deconstruct political trust, while gossiping or chatting with friends can construct political trust.

This study provides empirical evidence for understanding the relationship between unofficial media use and political trust, and also contains policy implications for constructing

political trust. (1) The development of unofficial media, while accelerating the realization of the needs of netizens for expressing interests, political participation, and power supervision, can also amplify social contradictions and widen the gap between netizens' psychological expectations and perception of reality, thereby deconstructing political trust. Therefore, it is necessary to strengthen the management of unofficial media, improve the access and approval mechanisms, establish information filtering mechanisms, and regulate the operation of unofficial media. It is important to establish a sound network public opinion monitoring, assessment, and handling mechanism to create a favorable public opinion environment and reduce the negative effects of negative information on netizens. (2) Boosting the subjective well-being of netizens is beneficial for curbing the deconstructive effect of unofficial media use on political trust. To achieve this, it is urgent to improve the management and construction capabilities of national institutions, provide high-quality public services to meet the interests and needs of netizens, and enhance their subjective well-being. Accelerating the construction of e-government, opening up multiple channels for expressing and providing feedback on interests, improving public participation mechanisms, and meeting the needs of netizens and national institutions for interaction, are also crucial. (3) Strengthening official media use can promote the constructive effect of subjective well-being on political trust. To achieve this, it is necessary to enhance the influence of official media, innovate the forms and content of official media communication, in order to reduce the crowding-out effect of unofficial media on official media; disclose information in a timely manner, to reduce misunderstandings of the state institutions caused by information asymmetry; enhance the ability to set public agendas in the online space, and seize the initiative in the dissemination of online content, to influence the political attitudes of netizens. (4) Improving media literacy among netizens can effectively alleviate the deconstructive effect of unofficial media use on political trust. To achieve this, it is necessary to reasonably and effectively allocate educational resources to improve the education level of netizens, enhance their ability to select information, discern right from wrong, and interpret information, and enable them to correctly perceive the gap between the virtual environment created by the media and real life, and adjust their psychological expectations and the gap between reality in a timely manner.

The paper has three limitations. First, the data in this study is non-tracking survey data, making it difficult to examine longitudinally the trends and characteristics of the impact of unofficial media use on trust in government over time. Second, although this study examined the mediating role of subjective well-being in unofficial media use on political trust, whether there are other mediating variables that play an important role between the two is yet to be followed up with in-depth research. Last, due to the research purpose, this article does not attempt to investigate the impact of unofficial media use on political trust in different countries.

Future research can first attempt to locate longitudinal survey data to examine how unofficial media use and political trust change over time, and can also attempt to conduct comparative analysis of similarities and differences in the relationship between unofficial media use and political trust across different countries. Next, searching for other important pathways to explore the underlying mechanisms between unofficial media and political trust. Final, looking for the suitable instrumental variables to solve endogenous problem.

## Supporting information

**S1 File.**
(XLSX)

## Author Contributions

**Data curation:** Qian Hu.

**Funding acquisition:** Yanping Pu.

**Methodology:** Qian Hu.

**Supervision:** Yanping Pu.

**Writing – original draft:** Qian Hu.

**Writing – review & editing:** Qian Hu, Yanping Pu.

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
