## [Decision Letter · Decision Letter 0]

23 Mar 2023

PONE-D-23-04787Using Unofficial Media, Less Trusting of Polity? Evidence from ChinaPLOS ONE

Dear Dr. Hu,

Thank you for submitting your manuscript to PLOS ONE. After careful consideration, we feel that it has merit but does not fully meet PLOS ONE’s publication criteria as it currently stands. Therefore, we invite you to submit a revised version of the manuscript that addresses the points raised during the review process.

We look forward to receiving your revised manuscript.

Kind regards,

Simon Grima, PhD

Academic Editor

PLOS ONE

Journal Requirements:

   "1.Financial support from National Social Science Fund of China (Project No.20XJY004) 

    2. Central University Basic Research Fund of China (Project No.2020CDJSK01TD02)" 

   "NO"

4. Please ensure that you include a title page within your main document. You should list all authors and all affiliations as per our author instructions and clearly indicate the corresponding author.

Reviewers' comments:

Reviewer's Responses to Questions

**Comments to the Author**

1. Is the manuscript technically sound, and do the data support the conclusions?

Reviewer #1: Partly

Reviewer #2: Yes

2. Has the statistical analysis been performed appropriately and rigorously? 

Reviewer #1: Yes

Reviewer #2: Yes

3. Have the authors made all data underlying the findings in their manuscript fully available?

Reviewer #1: Yes

Reviewer #2: Yes

4. Is the manuscript presented in an intelligible fashion and written in standard English?

Reviewer #1: Yes

Reviewer #2: Yes

5. Review Comments to the Author

Reviewer #1: 1. Due to the increasing influence of unofficial media in China, learning how to build political trust is crucial to advancing the country's system of national governance is the main point of discussion of this paper. But In the abstract section I am not clear with the findings and implications of the study.

2. And the study combines the bootstrap method to construct a mixed model with subjective well-being as intermediary variable and official media use as the moderating variable to empirically explore the impact of unofficial media use on political trust and its mechanism.

I would like to know the novelty of this research work.

3. In the introduction section please write something about why media is seen as playing a central role. Background of the study is missing. Include some discussion about China's media system as well. Discuss the impact of both mainstream and alternative media on popular perceptions of political philosophies.

4. The theoretical assumptions regarding how this official media use can be used as a moderating variable to empirically study the impact of unofficial media use on political trust and its mechanism leave me unsatisfied as well.

5. What new insights this manuscript adds to the existing literature?

6. It would be helpful if you could deepen the development of the theoretical model and provide additional literature for testing hypotheses.

7. Clarify the sample design and selection part.

8. This is a study in respect to China's background. What if we extrapolate these findings and apply them to other nations?

Reviewer #2: 1. I find the introduction insufficiently argued. the research gaps and study motivation should be clarified.

2.In the research methodology section, the authors are kindly asked to provide a short description of the adopted method of data analysis.

3. I would strongly suggest the author to add the discussion. It is critical to not just present results of research in conclusion but also describe what it means in the greater context. In addition, they should also discuss how their findings might generalize to other contexts.

6. PLOS authors have the option to publish the peer review history of their article (what does this mean?). If published, this will include your full peer review and any attached files.

Reviewer #1: **Yes: **KIRAN SOOD

Reviewer #2: **Yes: **Dr Sanjay Taneja

<quillbot-extension-portal></quillbot-extension-portal>

---

## [Author Response · Author response to Decision Letter 0]

17 May 2023

Dear Editor and Reviewers,

Hello! We received the review comments forwarded by the editorial office and we highly value them. We have carefully reviewed and discussed the comments. We sincerely appreciate the summary provided by the editor and the constructive suggestions made by the reviewers. These have helped to improve the clarity of the logic, rigor of the argument, and reliability of the conclusions in our article, as well as deepening our understanding of the relevant issues.

Here are our responses to the editor and the two reviewers. 

Thank you.

Edit:

1. Thank you for stating the following financial disclosure: 

We've checked your submission and before we can proceed, we need you to address the following issues:

1. A note on the role played by the funder in this study.

Response: We have explained the role of the funder in this study in the cover letter. The details are as follows, and editors can also view our cover letter submission.

(1). National Social Science Fund of China (Project No.20XJY004), the grant was awarded to Yanping Pu, who was mainly involved in the study design and analysis.

(2). Central University Basic Research Fund of China (Project No.2020CDJSK01TD02), the grant was awarded to Yanping Pu, who was mainly involved in the study design and analysis.

(3). Scientific Research Plan Projects of Shaanxi Education Department (Project No.21JK0148), the grant was awarded to Ran Gu, who had no role in study design, data collection and analysis, decision to publish, or preparation of the manuscript.

2.A note on conflicts of interest.

Response: We have set out in our cover letter that there is no conflict of interest between the co-authors. Details are below, and the editors can also view our submitted cover letter.

The authors have declared that no competing interests exist.

3.About the minimum data set.

Response: In accordance with the requirements of the journal PLOS ONE, we have uploaded the data as supporting files. Details are below, and the editors can also view our submitted cover letter Data availability statement.

Reviewer #1: 

1.Due to the increasing influence of unofficial media in China, learning how to build political trust is crucial to advancing the country's system of national governance is the main point of discussion of this paper. But In the abstract section I am not clear with the findings and implications of the study.

Response: Thank you very much for the questions raised by the reviewer. We have made revisions to the abstract, mainly focusing on the logic of ‘necessity of the research → research methods → research results → research implications’, with a focus on highlighting the research results and implications. The specific changes made can be found in the abstract section on page 2 of the revised manuscript.

2.And the study combines the bootstrap method to construct a mixed model with subjective well-being as intermediary variable and official media use as the moderating variable to empirically explore the impact of unofficial media use on political trust and its mechanism.I would like to know the novelty of this research work.

Response: The novelty of this research work lies in the following three aspects: 

Firstly, as mentioned in the first question raised by the reviewer, it is crucial to enhance government trust in order to promote the national governance system in the context of increasing use and influence of unofficial media. Therefore, this paper takes a media perspective and constructs and verifies the mechanism of the impact of unofficial media use on political trust through logical deduction, supplementing and expanding the research results on factors influencing political trust. 

Secondly, previous studies on the influence of political trust have mainly been explained from the perspectives of institutionalism and culturalism. This study attempts to find subjective well-being as a mediating variable that is outside the institutionalism and culturalism perspectives to explain the internal mechanism between unofficial media use and government trust, which expands the research content. In addition, unlike previous studies, this paper did not ‘isolate’ or ‘merge’ unofficial media use and official media use after empirically testing the mediating effect of subjective well-being on the impact of unofficial media usage on political trust. Instead, it introduces the concept of moderation effects to explore the moderating role of official media usage in the mediating effect of subjective well-being, and deeply reveals the underlying reasons behind the impact of unofficial media usage on political trust. 

Finally, based on the overall exploration of the mechanism of the impact of unofficial media use on political trust, this study delves into the horizontal dimension, analyzing in-depth the impact of unofficial media use on trust in different political institutions, as well as the impact of different types of unofficial media use on political trust. This has refined and expanded the research on political trust, providing empirical support for a comprehensive understanding of the impact of unofficial media use on political trust, and for the effective construction of political trust through relevant policy-making and institutional innovation.

3.In the introduction section please write something about why media is seen as playing a central role. Background of the study is missing. Include some discussion about China’s media system as well. Discuss the impact of both mainstream and alternative media on popular perceptions of political philosophies.

Response: Thank you for the suggestions provided by the reviewer. We fully agree with the reviewer’s opinion. Therefore, we have added content on the research background and why the media is considered to play a core role in the introduction part. We have placed some discussions about the Chinese media system and the views and impacts of mainstream and alternative media on the public’s political philosophy in the theoretical analysis and research hypothesis part. This is because the introduction part includes a lot of content, and placing the discussion of the Chinese media system in the introduction part would make the introduction too long and less readable, and the problem would not be focused. Therefore, we placed it in the theoretical analysis and research hypothesis part. The specific modifications can be found in the revised version on pages 3-5 for the introduction and page 6 for the theoretical analysis and research hypothesis sections.

4.The theoretical assumptions regarding how this official media use can be used as a moderating variable to empirically study the impact of unofficial media use on political trust and its mechanism leave me unsatisfied as well.

Response: We include official media use as a moderating variable in this study based on both theoretical and empirical considerations. In terms of theory, firstly, previous research on the relationship between media use and political trust mainly takes two forms: Form 1 focuses on treating the media as a whole and exploring the relationship between media use and political trust; Form 2 focuses on a specific type or specific media and explores the relationship between media use and political trust. Form 1 is prone to overlooking the differential effects that different types of media have on political trust. Therefore, this study adopts Form 2. However, within Form 2 studies, research on the relationship between unofficial media use and government trust is extremely limited, and basically treats official media as a control variable or does not consider it as a control variable. In fact, official media and unofficial media are relative to each other. If official media is not controlled for, what role does it play and what kind of media role does it assume in the process of unofficial media use affecting political trust through the mediating ‘bridge’ of subjective well-being? This is also a question worth exploring. If the mediating effect of subjective well-being exists and official media use plays a moderating role in subjective well-being’s influence on political trust, then it indicates that there is a complex mechanism of moderating mediation effects in the process of unofficial media use affecting political trust. Proposing such a complex mechanism hypothesis is meaningful, as it helps to comprehensively consider the contingency mechanism of mediation effects and thus more deeply reveal the functioning process of the entire influence mechanism. Through theoretical analysis and logical deduction, this article ultimately concludes that official media use has a moderating effect in the mediating process. The reasoning is roughly as follows: the framing effect emphasizes that the public’s judgment on a particular issue is influenced by the media (Entman, 1993). As the ‘mouthpiece’ of the Party and the government, official media has the mission of strictly supervising the government, spreading political ideology, designing top-level policies, and promoting positive propaganda to cultivate public political trust through advancing mainstream values and actively building the country’s image (Gilley, 2009; Wang & Jin, 2019). Therefore, in the pathway where subjective well-being influences political trust, official media use can promote the positive effect of subjective well-being on political trust, that is, there exists a regulating mediating effect in the process of unofficial media deconstructing political trust. Official media use will affect the impact of subjective well-being on political trust. Firstly, the stronger the use of official media, the stronger the positive effect of official media’s positive guidance on the public, which is easy to shape the public’s positive and optimistic attitude, make the public full of hope in life, and thus strengthen the impact effect of subjective well-being. Secondly, the weaker the use of official media, the more limited the positive effect of official media’s positive guidance on the public, which is easy to attribute the decrease of subjective well-being to the dereliction of duty of government agencies, and thus lower political trust.

In terms of empirical analysis, we strictly tested and verified the mediation and moderation effects. Eventually, we confirmed that subjective well-being plays a mediating role in the relationship between unofficial media use and political trust, and that official media use has a moderating effect on the impact of subjective well-being on political trust. Through the combination of theory and empirical analysis, we believe that treating official media use as a moderating variable can expand on existing literature and provide new insights for future research on the relationship between media use and political trust. We hope that the reviewers will take this into consideration.

5. What new insights this manuscript adds to the existing literature?

Response: This paper provides several new insights into the existing literature. First, it confirms that subjective well-being is an important channel through which unofficial media deconstruct political trust. Second, in the impact of unofficial media use on political trust, official media use has a positive moderating effect on the mediating role of subjective well-being. Third, unofficial media use has differential effects on trust in different political institutions, and different types of unofficial media use also have heterogeneous effects on political trust. Specifically, unofficial media use significantly deconstructs trust in the central government, courts, and police, and can also deconstruct trust in these institutions by reducing subjective well-being, but has no significant impact on trust in township governments. Weibo or online communities and overseas media can directly deconstruct political trust, as well as deconstruct it by reducing subjective well-being, while gossip or chatting with friends can both directly build political trust and build it with the help of subjective well-being. Subjective well-being is positively catalyzed by CCTV in the mediating role of Weibo or online communities and overseas media on political trust. Interestingly, Xinhua News Agency, People’s Daily, and Sina website do not have a moderating effect on the impact of subjective well-being on political trust. These new insights expand the content of the existing literature and provide some reference for future research on government trust.

6.It would be helpful if you could deepen the development of the theoretical model and provide additional literature for testing hypotheses.

Response: Thank you for the reviewer's suggestions. Through a careful review of the literature, we have reorganized the theoretical model, added relevant variables and literature, which have significantly improved the quality of the paper. The specific modifications can be found in the theoretical analysis and research hypotheses section on pages 5-9 and the discussion section on pages 23-25.

7.Clarify the sample design and selection part.

Response: Thank you for the suggestions, we have revised the sample design and selection section accordingly. Specifically, we have made changes to the data and measurement section, and added a model selection section. Please refer to page 10 and 13 for details.

8.This is a study in respect to China's background. What if we extrapolate these findings and apply them to other nations? 

Response: This study is based on China, aiming to examine the intrinsic relationship between the use of unofficial media and political trust in the Chinese context. These inferences may be applicable to countries with similar backgrounds to China, but some research conclusions may not be applicable, which requires further exploration and analysis to obtain. Therefore, when extrapolating research results from one country to another, it is necessary to proceed with caution and be aware of cultural, social, economic, political, and other differences between two or more countries. As far as this study is concerned, it is important to consider unique cultural, historical, and social factors that may affect research results in the Chinese context. China’s media background is completely different from that of the West. Due to the different political culture and media context, media is regarded as the fourth political power in western countries (Lewis et al., 2008), while in China, it is considered as a tool for political propaganda (Liu & Raine, 2016), plays an important role in maintaining the stability of the regime and strengthening social governance (Chen & Sun, 2019). The difference may lead to inconsistent or even contradictory results in the impact of unofficial media use on government trust. However, overall, our research results can also generate hypotheses for future scholars to use these findings and provide relevant information for future research in other contexts. At the same time, the research results also provide some reference value for countries with similar backgrounds to China. Although conducting similar research in other countries for comparison and contrasting results may be beneficial, due to the research purpose, time, energy, and resources, we temporarily defer comparing and analyzing the similarities and differences between other countries and China in the relationship between unofficial media use and political trust. We sincerely thank the reviewers for their constructive suggestions, which have provided new ideas for our future research. In subsequent studies, we will attempt to compare and analyze the relationship between unofficial media use and political trust in different countries, increasing the depth and thickness of the paper. We hope the reviewer will take the above situation into consideration.

In addition to the revisions suggested by the reviewers, we have also checked and revised the grammar, vocabulary, and phrasing of the manuscript.

Objectively speaking, after carefully revising the manuscript based on the suggestions of the reviewers, the language of the paper has become more concise, the logic clearer, and the methods more rigorous and scientific, greatly improving the quality of the article. We deeply admire the academic attitude and level of the reviewers, and once again sincerely thank them for their valuable feedback!

Reference

[1] Entman RM. Framing: Toward Clarification of A Fractured Paradigm. Journal of Communication, 1993;43(4):51-58.

[2] Gilley B. Holbig H. The Debate on Party Legitimacy in China: A Mixed Quantitative/Qualitative Analysis. Journal of contemporary China, 2009;18(59):339-358. 

[3] Wang H. Jin JB. Media Contact and Subjective Well-being: An Empirical Study with Political Trust as a Mediator Variable. Journalism Research, 2019;(7):1-15+120.(In Chinese)

[4] Lewis J. Williams A. Franklin B. A compromised fourth estate? UK news journalism, public relations and news sources. Journalism Studies, 2008;9(1),1-20.

[5] Liu H. Raine JW. Why is there less public trust in local government than in central government in China? International Journal of Public Administration, 2016; 39(4),258-269.

[6] Chen JN. Sun LY. Media Influence on Citizens’ Government Trust: A Cross-Sectional Data Analysis of China. International Journal of Public Administration, 2019;42(13):1122-1134.

Reviewer #2: 

1.I find the introduction insufficiently argued. the research gaps and study motivation should be clarified.

Response: Thank you for the suggestions provided by the reviewers. We have revised the introduction section accordingly. Specifically, in the introduction section, we have reviewed the relevant literature on media use and political trust, and in light of the growing influence of unofficial media in the media era, we have identified some gaps in the existing research in terms of research perspective, mechanism, and content. In order to address these gaps, the purpose of this paper is to explore the relationship between the use of unofficial media and political trust, and its mechanisms. Based on this, we will examine the impact of unofficial media use on trust in different political institutions and the influence of different types of unofficial media use on political trust. Ultimately, this study aims to provide a theoretical basis and empirical evidence for enhancing government trust and promoting the construction of national governance systems in the media era. At the same time, it also provides some reference value for countries with similar backgrounds to China. Based on the above logic, we have clarified the research gap and research motivation. You can refer to pages 3-5 of the revised manuscript for specific modifications to the introduction section.

2.In the research methodology section, the authors are kindly asked to provide a short description of the adopted method of data analysis.

Response: Thank you for the reviewers’ suggestions, which made us realize the inadequacy in introducing the data analysis methods in our writing. As a result, we have added a subsection on model selection, specifically introducing the bootstrap method and the propensity value matches. After revising, the clarity and readability of the article have also been improved. We would like to express our gratitude to the reviewers once again for their valuable feedback, and we will take this as a lesson for future writing to avoid similar mistakes. Please refer to page 13 of the revised manuscript for specific modifications to the model selection section.

3.I would strongly suggest the author to add the discussion. It is critical to not just present results of research in conclusion but also describe what it means in the greater context. In addition, they should also discuss how their findings might generalize to other contexts.

Response: Thank you for the suggestions from the reviewer, which made us realize the deficiencies in the conclusion and discussion section. Based on the suggestions, we made four modifications to this section. Firstly, we provided a more in-depth discussion of the empirical results, including the non-significant effect of unofficial media use on trust in township government, the discussion of how grapevine discussions or chatting with friends can build political trust, and the discussion of the non-significant moderating effect of Xinhua News Agency, People’s Daily, and Sina website between subjective well-being and political trust. Secondly, we reorganized and wrote the research conclusions, which were derived from four aspects: main effects, mediating effects, moderating effects, and heterogeneity. Thirdly, we added policy implications for building political trust. Based on the research conclusions, we discussed what each conclusion means in a broader context and how our findings can be applied to other situations. Finally, we discussed the limitations and prospects of this study. Specific modifications can be found in the discussion section on pages 22-25 and in the conclusion section on pages 25-27 of the revised manuscript.

In addition to the revisions suggested by the reviewers, we have also checked and revised the grammar, vocabulary, and phrasing of the manuscript.

We are deeply grateful for the valuable feedback provided by the reviewer, whose academic expertise and dedication to scholarly inquiry we greatly admire. By carefully considering the reviewer's feedback, we have gained a deeper understanding of the issues at hand, and the revised article is now more rigorous and logically coherent. Once again, we express our sincere gratitude for the reviewer's assistance.

---

## [Editor Report · Decision Letter 1]

30 May 2023

Using unofficial media, Less trusting of Chinese polity?

--An analysis based on the moderated mediation effect

PONE-D-23-04787R1

Dear Dr. Hu,

We’re pleased to inform you that your manuscript has been judged scientifically suitable for publication and will be formally accepted for publication once it meets all outstanding technical requirements.

Kind regards,

Simon Grima, PhD

Academic Editor

PLOS ONE

Additional Editor Comments (optional):

The authors satisfactorily addressed the issues highlighted by the reviewers

Reviewers' comments:

<quillbot-extension-portal></quillbot-extension-portal>

---

## [Editor Report · Acceptance letter]

1 Jun 2023

PONE-D-23-04787R1 

Using unofficial media, Less trusting of Chinese polity? --An analysis based on the moderated mediation effect 

Dear Dr. Hu:

I'm pleased to inform you that your manuscript has been deemed suitable for publication in PLOS ONE. Congratulations! Your manuscript is now with our production department. 

Kind regards, 

on behalf of

Professor Simon Grima 

Academic Editor

PLOS ONE